# JOSENet: A Joint Stream Embedding Network for Violence Detection in Surveillance Videos

## Abstract

Due to the ever-increasing availability of video surveillance cameras and the growing need for crime prevention, the violence detection task is attracting greater attention from the research community. With respect to other action recognition tasks, violence detection in surveillance videos shows additional issues, such as the presence of a significant variety of real fight scenes. Unfortunately, available datasets seem to be very small compared with other action recognition datasets. Moreover, in surveillance applications, people in the scenes always differ for each video and the background of the footage differs for each camera. Also, violent actions in real-life surveillance videos must be detected quickly to prevent unwanted consequences, thus models would definitely benefit from a reduction in memory usage and computational costs. Such problems make classical action recognition methods difficult to be adopted. To tackle all these issues, we introduce JOSENet, a novel self-supervised framework that provides outstanding performance for violence detection in surveillance videos. The proposed model receives two spatiotemporal video streams, i.e., RGB frames and optical flows, and involves a new regularized self-supervised learning approach for videos. JOSENet provides improved performance while requiring one-fourth of the number of frames per video segment and a reduced frame rate compared to state-of-the-art methods.

## 1 Introduction

Violence detection is one the most important and challenging sub-tasks of human action recognition Akti et al. (2019). Many violent events, like fighting, might arise from different situations and places (e.g., burglary, hate crimes, etc.) and it is rather difficult to detect them in an early stage to guarantee security Perez et al. (2019). Very few tools are available to detect and prevent violent actions. One of the most popular measures to increase public security is to adopt Closed-Circuit TeleVision (CCTV) video surveillance systems. However, CCTVs still require an enormous manual inspection, which is often affected by human fatigue that may jeopardize quick decisions and crime avoidance Xu et al. (2019); Sernani et al. (2021). A significant alternative solution to raise the level of public safety is represented by the development of deep learning methods for the automatic detection of violent actions Islam et al. (2021); Sernani et al. (2021); Sumon et al. (2020); Ullah et al. (2021). However, detecting violent scenes in surveillance videos entails several challenges, such as actors and backgrounds that may significantly differ among different videos, different lengths, or resource limitations due to real-time surveillance. Moreover, it is not easy to find available labeled datasets to effectively perform detection in a supervised fashion.

To address the above issues for the violence detection task, this work aims at introducing JOSENet, a novel joint stream embedding architecture involving a new efficient multimodal video stream network and a new self-supervised learning paradigm for video streams. In particular, the flow-gated network (FGN) Cheng et al. (2021) receives two video streams, a spatial RGB flow and a temporal optical flow, as shown in Fig. 1. The proposed method adopts a very small number of frames per segment and a low frame rate with respect to state-of-the-art solutions in order to optimize the benefit-cost ratio from a production point of view. This cost reduction may lead, however, to an unwanted side-effect that can significantly affect performance accuracy. In order to compensate for such a performance decrease while still reducing any overfitting, we initialize the network with

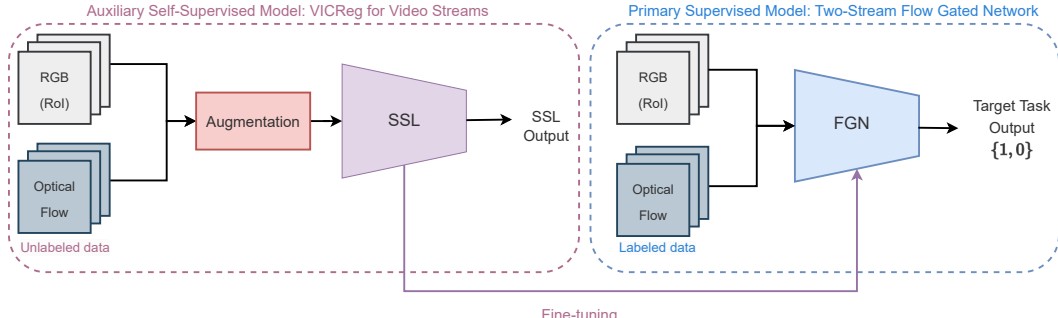

Figure 1: The proposed JOSENet framework. The primary target model (right) is tackled by using a novel efficient Flow Gated Network (FGN) which produces binary classification (1 if violence is detected, 0 otherwise) given optical flow and RGB segments. The FGN is pretrained by using a novel two-stream SSL method (left) that aims to solve an auxiliary task with unlabeled input data.

pretrained weights by involving self-supervised learning (SSL) approaches, which produce useful representations without relying on inputs annotated by humans. In particular, besides testing many different state-of-the-art SSL methods suitable for JOSENet, we propose a novel SSL algorithm specifically designed for video streams and based on the variance-invariance-covariance regularization (VICReg) Bardes et al. (2021). To the best of our knowledge, this is the first time a VICReg-like approach has been developed for a video stream architecture. The proposed SSL approach exploits the VICReg capabilities, thus, unlike other methods, it scales well with the dimension of the data, does not demand large memory, and prevents any collapse issue. The use of an SSL method makes JOSENet also robust to any lack of labeled data, which is often the case in real-life surveillance videos, and can improve the generalization capability of the model Yang et al. (2020).

We prove the effectiveness of the proposed JOSENet framework over the most popular datasets of violence detection under several conditions, highlighting the advantages and drawbacks of our method. Results show that the proposed JOSENet model is able to efficiently reduce both the number of frames per video and the frame rate while outperforming existing solutions.

## 2 RELATED WORK

**Violence Detection.**    In the last years, the analysis of violent actions has become tractable thanks to deep neural networks. An early work employed a VGG16 for optical flows Mukherjee et al. (2017). A modified Xception CNN is used in Akti et al. (2019) together with a Bi-LSTM to learn the long-term dependency. Investigation on the use of bidirectional temporal encodings can be found in Hanson et al. (2019). Besides introducing the CCTV-Fight dataset, 2D CNN VGG16 architectures were proposed in Perez et al. (2019) . A framework with localization and recognition branches was proposed in Xu et al. (2019). Alternative approaches are based on the 3D skeleton point clouds, e.g., extracted from videos via a pose detection module Su et al. (2020); Garcia-Cobo & SanMiguel (2023). In Sumon et al. (2020), several pretraining strategies were explored. Efficient spatio-temporal architectures were proposed in Islam et al. (2021); Kang et al. (2021).

**Self-Supervised Learning.**    Self-supervised learning (SSL) aims at learning representations from unlabeled data, and building generalized models. The first SSL approaches were proposed for spatial context prediction Doersch et al. (2016) and Jigsaw puzzle solution Noroozi & Favaro (2017). The contrastive learning (CL) approach Dosovitskiy et al. (2015) discriminates between a set of augmented labels. The contrastive predictive coding extracts Oord et al. (2019) useful representations from high-dimensional data. In Haresamudram et al. (2020), CL was applied to human activity recognition. While CL obtained competitive performance with respect to supervised representation Chen et al. (2020); He et al. (2020), it does not scale well with the dimension of the data, and it tends to require large memory demands. Regularized methods solve these problems Grill et al. (2020); Caron et al. (2021). In particular, the Barlow Twins method Zbontar et al. (2021) naturally avoids collapse by the measure of cross-correlation matrix between the two outputs of a Siamese neural network. Inspired by Zbontar et al. (2021), the Variance-Invariance-Covariance Regularization (VICReg) Bardes et al. (2021) has been proposed for multimodal data, showing good scaling ability and limited memory demand. A new variant of VICReg for video streams is proposed for JOSENet.

**Self-Supervised Learning for Video**    Several SSL techniques were specifically proposed for video streams, starting from Misra et al. (2016); Fernando et al. (2017); Lee et al. (2017). One of the first works acting on a two-stream architecture was presented in Taha et al. (2018). Spatiotemporal 3D CNNs were introduced in SSL by Kim et al. (2019). In Wang et al. (2019), a task was defined to predict numerical labels. A method based on contrastive predictive coding for video representation learning was developed in Lorre et al. (2020). Dense predictive coding was proposed for learning spatiotemporal video embeddings Han et al. (2019). Differently, CoCLR Han et al. (2020) exploits complementary data (i.e., optical flow) as additional positive samples in a new co-training regime. The pretext-contrastive learning (PCL) Tao et al. (2021) is a joint optimization framework for both CT and pretext tasks. The only work that uses SSL for the violence detection task introduces an iterative learning framework based on two experts feeding data to each other where the SSL expert is a C3D network Degardin & Proenca (2020). The classification network involved in the proposed JOSENet framework is an advanced version of the C3D. Also, in Seo et al. (2022), an SSL approach is adopted to pretrain a module for selecting informative frames for abnormal action recognition. Our aim instead, is to obtain a better-performing network without using additional modules that could slow down the inference speed, which is critical for violence detection.

## 3    Proposed method

The proposed JOSENet framework for violence detection is basically composed of two parts, as depicted in Fig. 1: a primary target model and an auxiliary SSL model. The target part is composed of the two-stream flow gated network, involving both spatial and temporal flows, which performs violent action detection using labeled data. The two-stream architecture guarantees significant performance. To reduce both memory and computational costs, we use a lightweight setting for the model in terms of the number of frames per video. The JOSENet framework benefits from an auxiliary model to avoid any loss of performance. This auxiliary network implements a novel SSL method receiving unlabeled data as input. The weights of the auxiliary model are optimized by minimizing the VICReg loss, and subsequently employed in the pretraining of the primary supervised model. Following the pretraining phase, a fine-tuning strategy is utilized to refine and tailor the pretrained weights to the specific requirements of the primary task. The auxiliary SSL network allows JOSENet to achieve the best trade-off between performance and employed resources. In the following, we focus in detail on the two models of the proposed framework.

### 3.1    Primary Model: An Efficient Two-Stream Flow Gated Network

The primary target model is based on a two-stream flow gated network (FGN). The choice of a two-stream architecture is motivated by the benefits brought by multimodal video architectures and by the excellent performance that the FGN achieved on the RWF-2000 dataset Cheng et al. (2021). This network consists of three modules: a spatial block, a temporal block, and a merging block.

**Spatial Block.** The spatial RGB module receives as input consecutive frames that are cropped to extract the region of interest (ROI). The ROI aims to reduce the amount of input video data, making the network focus only on the area with larger motion intensity. The computation of the ROI involves normalization and a subtraction of the mean of each optical flow frame for denoising purposes. Given the normalized and denoised optical flow frame $S_i$, the magnitude can be computed as $\sqrt{S_{i,x}^2 + S_{i,y}^2}$ where $S_{i,j}$ represents the $j$-th component of the $i$-th frame. The sum of the magnitudes of each frame produces a $224 \times 224$ motion intensity map, on which the mean is computed and used as a threshold to additionally filter out the noise (i.e., zeroing out the motion intensity map values if less than this threshold). To obtain the center of the ROI based on the motion intensity map a probability density function along the two dimensions $x, y$ of the motion intensity map. Ten different candidates are selected to be the center of the ROI, and are extracted randomly from this probability density function. The final value of the center $(c_x, c_y)$ is obtained by the average of these 10 points for better robustness. The ROI is extracted by a patch of size $112 \times 112$ from the RGB frames based on $(c_x, c_y)$, thus a cubic interpolation is applied to reconstruct $N$ frames with size $224 \times 224$. Once processed, the output of the RGB block passes through a ReLu activation function. The resulting input dimension of the RGB block is $3 \times N \times 224 \times 224$, where the first dimension represents the RGB channels of the video segment.

**Temporal Block.** The temporal block receives as input the optical flow of the sequence of frames. The flow is computed by using the Gunnar Farneback's algorithm, as in Cheng et al. (2021). For each RGB segment $(F_0, F_1, \ldots, F_s)$ of $N$ frames of size $224 \times 224$, an optical flow frame is computed from each couple $(F_{i-1}, F_i)$. The resulting dimension of the flow block is $2 \times N \times 224 \times 224$. A sigmoid activation function at the end of this branch scales the output for the RGB embeddings.

**Merging Block.** The last FGN module defines the fusion strategy that manages both the RGB and the flow streams. In particular, the output of the RGB block and the flow block are multiplied together and processed by a temporal max pooling. This is a self-learned pooling strategy that utilizes the flow block as a gate, aiming to decide which information from the RGB block should be maintained or dropped. Finally, the fully-connected layers generate the output for that input.

**Computational Enhancement.** Since violence detection is a real-time application, it is necessary to reduce the computational cost as much as possible, thus finding more efficient ways to produce inference Xu et al. (2019), while deploying the model with reduced memory usage and frame rate during inference. To maintain the cost as low as possible, it is necessary to first deal with the number of frames $N$ in the input video segment while, at the same time, the network should be able to learn the correct features by using a correct window size $T_{\text{tf}}$ (temporal footprint). The vanilla FGN Cheng et al. (2021) uses $N = 64$ frames with $N_s = 12.8$ frames per second (FPS). Instead, to speed up the inference, we use $N = 16$ which is a common value for the most used action recognition architectures such as R(2+1)D Tran et al. (2018), C3D Tran et al. (2015), I3D Carreira & Zisserman (2018) and P3D Qiu et al. (2017). In action recognition, it is known that a very short window can lead to perfect recognition of most activities Banos et al. (2014) while at the same time, the classification performance generally increases by using a very high frame rate. However, as pointed out in Harjanto et al. (2016), action recognition methods do not always obtain their best performance at higher frame rates, but the best results are achieved by each method at different frame rates. In this way, we treat it like a hyperparameter that we aim to reduce for practical reasons. Thus, by experimenting with different frame rates we find that a value of $N_s = 7.5$ is an optimal trade-off between computational cost and performances, obtaining as a result a temporal footprint of $T_{\text{tf}} = \frac{N}{N_s} = 2.13$s. This choice can be considered appropriate for the broad category of violent actions. Further generalization performance can be found in Section A.6 of the appendix.

**Efficient Implementation.** To reduce the segment length $N$ we modify the $2 \times 2 \times 2$ max-pooling layers of the merging block into a $1 \times 2 \times 2$ (i.e., reducing by 1 the temporal dimension) so that the output dimension for that block is unchanged. In addition, to approximately halve the memory requirements while speeding up arithmetic, we use mixed-precision Micikevicius et al. (2018). With the aim of reducing the internal covariance shift, a 3D batch-norm layer is applied after each activation function of all the blocks of our architecture, except for the fully connected (FC) layers. Lastly, to avoid overfitting, spatial dropout with $p = 0.2$ is applied after each batch normalization layer of the first two 3D convolutional blocks, in both RGB and optical flow blocks. Details on the parameters and layers of our FGN can be found in Section A.2. The proposed model achieves a significant reduction of computational complexity compared to the original network in Cheng et al. (2021). Specifically, our model required only 4.432G multiply-accumulate operations (MACs), whereas the original architecture demanded 33.106G MACs, indicating a **7-fold reduction in computational load**. In addition, through a 4x smaller segment size and a reduced temporal footprint, we are able to significantly **reduce the memory requirements by 75%** and achieve a **two-fold increase in signaling alarm speed in real-life scenarios**. This result highlights the potential of our framework for efficient and effective neural network design.

## 3.2 Auxiliary Model: VICReg for Joint Video Stream Architectures

**Self-Supervised Pretraining.** The original FGN Cheng et al. (2021) does not involve any pretraining. In this work instead, we investigate the use of self-supervised pretraining as opposed to fine-tuning approaches 1) to compensate for the performance loss due to the resource limitation of the primary network, 2) to deal with unlabeled data typical of real-world surveillance video applications, and 3) to improve the generalization performance by avoiding a bias toward the source labels on the source task Yang et al. (2020). We implemented four different SSL techniques: odd-one-out (O3D) Fernando et al. (2017), arrow of time (AoT) Wei et al. (2018), space-time cubic puzzle (STCP) Kim et al. (2019), and VICReg Bardes et al. (2021). Clearly, all the above SSL techniques can be adapted to our FGN. Further details can be found in Section A.3 of the appendix.

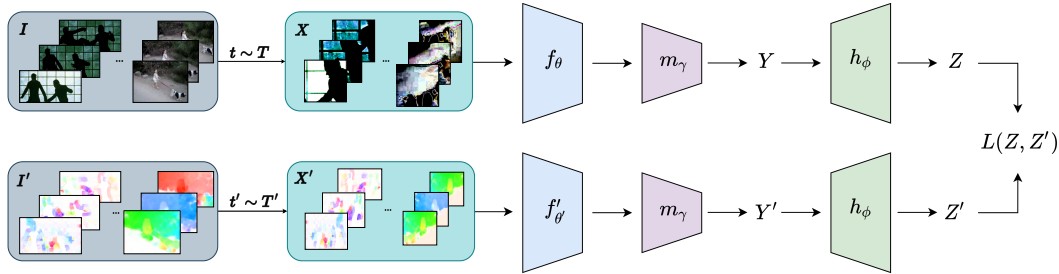

Figure 2: The proposed VICReg solution for the JOSENet framework. $I$ and $I'$ are respectively a batch of RGB and flow segments that are transformed through data augmentation into two different views $X$ and $X'$. In particular, a strong random cropping strategy and some other augmentation techniques are applied to RGB frames while the flow frames are only flipped horizontally. The RGB branch is represented by $f_\theta$, the optical flow branch is $f'_{\theta'}$, $m_\gamma$ is the merging block without the temporal max pooling and finally, the $h_\phi$ is the expander as in the VICReg original implementation. The VICReg loss function $L(Z, Z')$ is computed on the embeddings $Z$ and $Z'$.

**Proposed VICReg for JOSENet.** In our framework, we focus on the variance-invariance-covariance regularization (VICReg) Bardes et al. (2021), an SSL method for joint embedding architectures that preserves the information content of the embeddings, while not demanding large memory requirements, contrastive samples nor memory banks, unlike what happens in contrastive methods Jing et al. (2022). To the best of our knowledge, for the first time VICReg is applied to video streams. The proposed VICReg solution for JOSENet relies on the joint information of the augmented RGB and flow batches and it is depicted in Fig. 2.

Let us consider an RGB batch $I$ and an optical flow batch $I'$ both related to the same input segment. Two augmented views of these batch $X$ and $X'$ can be produced by using random transformations $t$ and $t'$ sampled from a distribution $T$. The augmented batches are fed into two different encoders $f_\theta$ and $f'_{\theta'}$. The two branches do not have the same architectures and do not share the same weights. The output of these branches is fed as input into two Siamese merging blocks $m_\gamma$ that have the same architecture as the merging block but without the temporal max pooling. Indeed, the removal of this pooling seems to be beneficial thanks to the increase of the expander input dimensionality. The output representations $Y$ and $Y'$ are fed as input into two Siamese expanders $h_\phi$ which produces batches of embeddings $Z = h_\phi(Y)$ and $Z' = h_\phi(Y')$ of $n$ vectors of dimension $d$. By utilizing the Siamese merging block during the self-supervised phase, we are able to generate representations of the input data by leveraging a significant portion of the FGN architecture. As a result of this approach, we are able to obtain highly informative and useful feature representations of the input data, which can subsequently be utilized for the primary task. This particular solution would be infeasible with other SSL techniques and can be designed only by using a method that can handle multimodality, such as VICReg. Indeed, among the several configurations tested, we will demonstrate that this is the best setup for pretraining our two-stream FGN. We leverage the basic idea of VICReg to use a loss function with three different terms. The variance regularization term $v(Z)$ is computed along the batch dimension as the standard deviation of the embeddings and aims to prevent a complete collapse. The covariance regularization term $c(Z)$ encourages the network to decorrelate the dimensions of the embeddings so that similar information is not encoded. This kind of decorrelation at the embedding level leads to a decorrelation at the representation level as well. The invariance term $s(Z, Z')$ is the mean-squared Euclidean distance between each pair of embedding vectors. The overall loss function $L(Z, Z')$ is a weighted average of these terms:

$$L(Z, Z') = \lambda s(Z, Z') + \mu[v(Z) + v(Z')] + \nu[c(Z) + c(Z')] \tag{1}$$

where $\lambda, \mu$ and $\nu$ are hyperparameters (specifically, $\lambda = \mu = 25$ and $\nu = 1$ works best in most of the contexts). Since VICReg is an information maximization method, it does not require the use of techniques generally used in contrastive methods. Moreover, although it was proposed for a Siamese network Bardes et al. (2021), one of its greatest advantages is that the two branches could also not share the same parameters, architectures, and more importantly input modality.

**Configuration.** In Bardes et al. (2021), the input size of the expanders $h_\phi$ is set to 2048. In our case, using a merging block $m_\gamma$ with an unchanged structure would produce an input of size 128. Since it is paramount to include most of the two-stream architecture without reducing too much the expander dimensionality, we remove the temporal max-pooling in the merging block. In this way,

the expander input dimensionality grows from 128 to 1024, finding a good trade-off between the two constraints. Thus, the output representations $Y$ and $Y'$ have dimension 1024 and are fed as input into the two Siamese expanders $h_\phi$. The expanders have the same structure as the original method: 3 FC layers of size 8192, where the first two layers utilize batch normalization and ReLU.

## 4 EXPERIMENTAL RESULTS

### 4.1 EXPERIMENTAL SETTINGS

**Datasets.** To train and validate the model for supervised learning we use the RWF-2000 dataset Cheng et al. (2021), involving 2,000 heterogeneous videos, 5 seconds long, and captured at $N_s = 30$ FPS by real-world surveillance cameras. In line with the existing literature, we also use HMDB51 Kuehne et al. (2011) (51 classes spread over 6,766 clips) and UCF101 Soomro et al. (2012) (nearly twice the size of HDMB51) datasets. Although a larger dataset can be beneficial in most cases, SSL techniques generally outperform transfer learning when the amount of pretraining is small Yang et al. (2020). Moreover, since a strong domain similarity can be very useful Yang et al. (2020), we use the UCF-Crime Sultani et al. (2019) as an additional dataset. We maintain the default train-test split in all the datasets used in both target and auxiliary tasks. The number of segments for the training and validation sets of each dataset is summarized in the appendix, where further dataset analysis can be found as well.

**Preprocessing Methods.** We use the same frame resolution as in Cheng et al. (2021) ($224 \times 224$) with each video segment generated with $N_s = 7.5$ FPS in a sliding window manner Banos et al. (2014). Details on the preprocessing methods for each auxiliary task can be found in the appendix.

**Data Augmentation.** In the target task, we apply color jitter to RGB frames and random flip to both RGB and flow frames. On the other hand, for each auxiliary task, we use different kinds of augmentations, as defined in Subsection 3.2. For the proposed VICReg, differently from Bardes et al. (2021); Chen et al. (2020), we use a stronger random cropping strategy in the RGB augmentation pipeline, by imposing a scaling factor within the range $[0.08, 0.1]$, which is confirmed to be more efficient from our tests. We call this augmentation the "zoom crop" strategy. For what concerns the flow segments, a simple horizontal flip (with 50% probability) is applied. A visualization of the VICReg video input can be seen in Fig. 2. Further details can be found in the appendix.

**Metrics.** Similarly to Sernani et al. (2021), we evaluate the performance by the following metrics: the accuracy, F1-score, true negative rate (TNR), true positive rate (TPR), and also the ROC curve and the area under the curve (AUC) to understand the model diagnostic capability in identifying violent videos.

### 4.2 EXPERIMENTAL RESULTS WITHOUT PRETRAINING

The first experiments aim to find the best baseline model without self-supervised pretraining. We first tackle the hyperparameter tuning from which we found our best parameters for the network. More specifically, we train the network on 32 batches for a total of 30 epochs together with an early stopping procedure with a patience of 15 epochs. The number of frames for each segment is $N = 16$, sampled at $N_s = 7.5$ FPS. We use a $p = 0.2$ dropout probability for the classification block. A binary cross-entropy loss is employed with a stochastic gradient descent (SGD) optimizer (momentum 0.9 and 1e-6 weight decay), as it is a state-of-the-art choice for violent detection Cheng et al. (2021). A cosine annealing scheduler is implemented, which starts from the initial learning rate value of 0.01, and decreases for 30 epochs to reach a minimum of 0.001. With these settings, we obtain 84.25% F1-score. To avoid overfitting at this stage, we add spatial dropout with a probability of 0.2. The benefit of the spatial dropout can be seen in the appendix. In this case, we obtain 85.87% F1-score, thus we increase it by +1.62%, 85.5% TNR, 86.25% TPR, and 0.924 AUC. We use this model as a "baseline" for comparisons.

### 4.3 EXPERIMENTAL RESULTS WITH SELF-SUPERVISED LEARNING

Now we evaluate the target task using the embeddings obtained with SSL pretraining. In particular, for each technique, we pretrain on three different datasets: HMDB51, UCF101, and UCF-Crime.

Table 1: Results obtained by pretraining our FGN using the non-regularized SSL methods and fine-tuning it on the target task. Each SSL method is tested with a different dataset for pretraining.

| SSL | Dataset | Accuracy | F1 | TNR | TPR | AUC |
|---|---|---|---|---|---|---|
| O3D Fernando et al. (2017) | HMDB51 | 82.62 | 82.62 | 84 | 81.25 | 0.897 |
| O3D Fernando et al. (2017) | UCF101 | 82.75 | 82.74 | 82.25 | 83.25 | 0.888 |
| O3D Fernando et al. (2017) | UCF-Crime | 83.37 | 83.37 | 83.75 | 83 | 0.901 |
| AoT Wei et al. (2018) | HMDB51 | 84.75 | 84.74 | 86.25 | 83.25 | 0.911 |
| AoT Wei et al. (2018) | UCF101 | 84.5 | 84.49 | 82.75 | 86.25 | 0.911 |
| AoT Wei et al. (2018) | UCF-Crime | 84 | 83.97 | 80 | 88 | 0.900 |
| STCP Kim et al. (2019) | HMDB51 | - | - | - | - | - |
| **STCP** Kim et al. (2019) | **UCF101** | **86.25** | **86.25** | 85.25 | **87.25** | **0.913** |
| STCP Kim et al. (2019) | UCF-Crime | 85.25 | 85.23 | **88.75** | 81.75 | 0.912 |

During pretraining we use the same hyperparameters of the target task with the difference in weight decay value of 1e-6 while epochs and batch size vary based on the technique used. In each trial, we maintain the maximum number of iterations of the cosine annealing scheduler equal to the number of epochs. While for VICReg we use the custom loss described by eq. 1, in all the other techniques a cross-entropy loss is applied. A fine-tuning strategy is applied by training the primary model on the target task.

**JOSENet with non-regularized SSL approaches.** Results for JOSENet with non-regularized SSL methods are shown in Table 1. As expected, the O3D technique does not provide useful embeddings for the target task. In fact, the results are all worse than the baseline model by a wide margin. It is useful to note that the results are better in UCF-Crime, while HMDB51 produces the worst pretraining weights. For what concerns the Arrow of Time (AoT) method, the results are unsatisfactory, probably because this technique (like the O3D one) is strongly dependent on the architecture chosen (in both cases they rely on 2D CNNs). It is interesting to see that the HMDB51 and UCF101 obtain similar results while UCF-Crime produces the worst performance. This may due to a trivial learning problem due to low-level cues. This is also confirmed by the high accuracy obtained on the auxiliary task itself. For STCP some improvements are achieved by using the UCF101 dataset. It is important to point out that we obtained a slight improvement compared to the baseline model and higher results compared to the other SSL pretraining. Differently from the previous methods, the STCP is built to be used on 3D CNNs, confirming that the architecture similarity plays an important role in the choice of the correct non-regularized SSL technique used.

**JOSENet with Pretraining Dataset Selection for VICReg.** We focus the attention on an approach without shared weights by pretraining simultaneously both RGB and optical flow branches. We assume that the best configuration involves a zoom crop strategy without considering the temporal max-pooling in the merging block. To validate this hypothesis and understand the best dataset for pretraining we test on the remaining two datasets: UCF101 and UCF-Crime. To provide a fair comparison between HDMB51 and the remaining ones, while maintaining a feasible training time (i.e., each pretraining on the HMDB51 dataset lasts around 12 hours), in this phase we pretrain on a portion of UCF101 and UCF-Crime with 16 batches by random sampling 15% of each of the datasets. The results are shown in Table 2. As we can see, VICReg pretrained on UCF101 outperforms the best model obtained so far. Compared to the baseline, we show an increase in F1-score (+0.5%), TNR (+0.25%), and TPR (+0.75%). Similarly, the model pretrained on UCF-Crime and tested on the target task shows a huge increase in performance in terms of F1-score (+0.87 %) and TPR (+2.75 %). This result underscores the importance of the quality of the dataset for the auxiliary task. Moreover, as expected, the dataset domain similarity between auxiliary and target tasks seems to be beneficial, resulting in an important factor to be considered when choosing the pretraining dataset.

**Final Results for JOSENet.** To obtain our best performances and to further validate our solution, we train our best configuration on the entire UCF-Crime dataset using an NVIDIA Tesla V100S GPU with 32GB GPU RAM and CPU Intel Xeon Gold 6226R CPU @ 2.90GHz with 15 cores, thus increasing the batch size from 16 to 64. In this case, we directly drop the merging block pretrained weights. This behavior suggests that the merging block weights (like the expander ones) are very important only during the pretraining phase. In other words, the merging block in VICReg helps the RGB and flow branches to learn the correct features but, with the increase of the batch size, the role of the merging block weights remains fundamental in VICReg while losing its relevance in the target task. The final results shown in Table 2, meet our expectations. We reach a **F1-score of 86.5%**

Table 2: Results obtained by our proposed VICReg method on a random 15% subset of the datasets UCF101 and UCF-Crime. In particular, UCF-Crime surpasses by a wide margin UCF101 on TPR and more importantly on F1-score. The final results obtained with a pretraining on the entire UCF-Crime dataset are shown in the last row.

| Dataset | Accuracy | F1 | TNR | TPR | AUC |
|---|---|---|---|---|---|
| HMDB51 | 85.37 | 85.37 | 84 | 86.75 | 0.924 |
| UCF101 | 86.37 | 86.37 | 85.75 | 87 | 0.922 |
| UCF-Crime | **86.75** | **86.74** | 84.5 | **89** | 0.918 |
| UCF-Crime (100%) | 86.5 | 86.5 | **88** | 85 | **0.924** |

Table 3: Representations from InfoNCE, UberNCE, CoCLR and JOSENet are evaluated on the target task, with a pretraining obtained on a random 15% subset of UCF-101. The ✗ indicates the random initialization of the branch during target training. In the last rows, we simultaneously initialize both the RGB and flow blocks of the FGN model.

| Method | RGB | Flow | Accuracy | F1-score | TNR | TPR | AUC |
|---|---|---|---|---|---|---|---|
| InfoNCE Oord et al. (2019) | ✓ | ✗ | 85.12 | 85.12 | 84 | 86 | 0.906 |
| UberNCE Han et al. (2020) | ✓ | ✗ | 84.5 | 84.5 | 77 | **92** | 0.911 |
| CoCLR Han et al. (2020) | ✓ | ✗ | 84.25 | 84.25 | 84 | 84 | 90.86 |
| JOSENet (ours) | ✓ | ✗ | 84.62 | 84.62 | 85.75 | 83.5 | 0.917 |
| InfoNCE Oord et al. (2019) | ✗ | ✓ | 85.12 | 85.12 | 87 | 83 | 0.907 |
| UberNCE Han et al. (2020) | ✗ | ✓ | 84 | 83.99 | 86 | 82 | 0.905 |
| CoCLR Han et al. (2020) | ✗ | ✓ | 84.25 | 84.25 | 84 | 84 | 0.908 |
| JOSENet (ours) | ✗ | ✓ | 84.49 | 84.49 | 82.75 | 86.75 | 0.900 |
| InfoNCE Oord et al. (2019) | ✓ | ✓ | 83.62 | 83.59 | 89 | 79 | 0.897 |
| UberNCE Han et al. (2020) | ✓ | ✓ | 83.87 | 83.80 | **91** | 77 | 0.895 |
| CoCLR Han et al. (2020) | ✓ | ✓ | 83.5 | 83.44 | 90 | 78 | 0.898 |
| JOSENet (ours) | ✓ | ✓ | **86.4** | **86.4** | 85.7 | 87 | **0.922** |

**(+0.63%)** which is a solid improvement compared to the baseline model. The confusion matrix (shown in appendix) indicates that the model reaches **88% (+2.5%) in TNR** and **85% (-1.25%) in TPR**. While a decrease in the TPR rate is acceptable, the increase in TNR is well received. In fact, from an application point of view, the TNR is an important metric that avoids stressing out the user with excessive false positives, given that the number of non-violence samples would be greater than the violent ones in a real case scenario. Furthermore, by reducing false positives, we can mitigate the risk of wrongful actions that could be taken against individuals who are mistakenly identified as violent. The **AUC reaches a value of 0.924** which is similar to the baseline model, demonstrating a good capability to distinguish between classes. Although JOSENet performs slightly worse than the state of the art for RWF-2000 Cheng et al. (2021), it clearly represents a more efficient and faster solution thanks to a four-time smaller segment length (16 instead of 64) and a smaller FPS required (7.5 instead of 12.8). This proves that the proposed framework features an excellent compromise between performance and efficiency.

**State-of-the-Art Comparison.** In this section, we compare JOSENet with previous SOTA self-supervised approaches: InfoNCE Oord et al. (2019), UberNCE Han et al. (2020) and CoCLR Han et al. (2020). We decide to take as reference the results obtained on the 15% subset of UCF-101 with JOSENet. For a fair comparison, we pretrain on the same subset either RGB or flow blocks using the SOTA methods by scaling down accordingly some of their hyperparameters (see appendix). Then, as usual, the obtained weights are used as pretraining for the target task. The results are shown in Table 3. We can observe that InfoNCE surpasses both the accuracy and F1-score performance of JOSENet and all the other methods when only the RGB or the flow blocks is pretrained. However, the AUC results show that UberNCE is still a valid alternative to the instance-based SSL in such a situation. As expected, when two pre-trained branches are used, CoCLR has better AUC performance compared to all the other SOTA methods thanks mainly to the co-training scheme. Nevertheless, a huge drop in performance happens in the SOTA methods compared to JOSENet, which obtains the best results on this setting: +1.28% accuracy and F1-score, with +0.011 AUC from the second best method. These results can be explained by the fact that all the SOTA methods do not take into consideration the merging block of the FGN while JOSENet is able to exploit that part of the architecture during pretraining, thus producing better embeddings for the target task.

Table 4: Results of the target task achieved through pretraining with the VICReg method on HMDB51 dataset, utilizing two Siamese branches, either RGB or flow. The ✗ indicates the random initialization of the branch during target training. In the last row, we used simultaneously the previously pretrained Siamese models on the target task.

| RGB | Flow | Accuracy | F1 | TNR | TPR | AUC |
|-----|------|----------|-------|-------|-------|-------|
| ✓ | ✗ | 84.62 | 84.62 | 85.75 | 83.5 | 0.917 |
| ✗ | ✓ | 84.5 | 84.49 | 82.07 | 87.03 | 0.918 |
| ✓ | ✓ | 84.37 | 84.37 | 82.75 | 86.75 | 0.900 |

Table 5: Results obtained on the target task by pretraining the VICReg on HMDB51 using different configurations. In particular, we test our model by including or removing the zoom crop (ZC) augmentation strategy and/or the temporal pooling (TP) in the merging block.

| ZC | TP | Accuracy | F1-score | TNR | TPR | AUC |
|----|-----|----------|----------|-------|-------|--------|
| ✓ | ✗ | **85.37** | **85.37** | 84.01 | **86.75** | **0.924** |
| ✓ | ✓ | 82.5 | 82.49 | 84.75 | 80.25 | 0.8927 |
| ✗ | ✗ | 85.25 | 85.22 | **90.15** | 81.57 | 0.9159 |

## 5 ABLATION STUDIES

**Siamese Architecture.** With the aim to understand if a simpler Siamese architecture can be sufficient for good pretraining, we pretrain either RGB or flow branches with shared weights. In both cases, we pretrain the branches on HMDB51. During the target task training, the non-pretrained block is randomly initialized. As an additional experiment, to avoid this random initialization, the resulting pretrained blocks are used simultaneously in the target task. The results are shown in Table 4. All these approaches seem to be inefficient, confirming that in order to improve the embeddings of both RGB and flow branches, JOSENet needs to use the complementary information provided by both RGB and flow views during the pretraining phase.

**Augmentation Strategies.** The different augmentation strategies are shown in Table 5. We first pretrain our model by using the zoom crop strategy. At the same time, we try to avoid an excessive reduction in the dimensionality of the expander input by removing the temporal max-pooling in the merging block. While the results on the F1-score are slightly lower than our baseline, we obtain an increase in TPR by +0.50% and a slight increase in AUC suggesting a feasible model configuration. To find a confirmation of this approach, using the zoom crop strategy, we apply the temporal pooling in the merging block, obtaining on the target task a very low value for most of the evaluation metrics used. This test shows that some issues occur when there is a very small bottleneck between the encoder and the expander. In fact, the input expander dimensionality is reduced from 1024 to 128. Successively, the network is pretrained with the random crop Bardes et al. (2021) while avoiding the temporal pooling in the merging block. The results are not optimal and suggest that zoom crop augmentation is crucial for the network to extrapolate the most useful features.

## 6 CONCLUSION AND FUTURE WORK

In this work, we introduced JOSENet, a novel regularized SSL framework involving a modified VICReg for a two-stream video architecture. The proposed framework is able to tackle effectively the violence detection task, a challenging research topic in computer vision. The proposed JOSENet framework has proven to achieve outstanding performance for violence detection, while maintaining solid generalization capability, as it is able also to detect non-violent actions. In the future, it would be interesting to focus on possible limitations, e.g., avoiding any possible bias in unfair prediction, reducing the risk of false negatives, assessing the robustness against real-world issues on data (e.g., occlusions, light conditions), and improving the efficiency of the optical flow branch. Moreover, different models (e.g., Islam et al. (2021); Garcia-Cobo & SanMiguel (2023)) could be redefined and compared according to the proposed JOSENet framework.

## STATEMENTS

**Ethics.** We acknowledge the importance of the ICLR Code of Ethics and ensure that this work follows it. We recognize that the application of violence detection carries ethical responsibilities. For this reason, we prioritized metrics (i.e., TNR) that reduce the false positive rate which could cause innocent individuals to be wrongly accused or flagged as potential threats. Moreover, to avoid possible bias and discrimination we have used datasets that are publicly available ensuring the broadest possible range of data sources.

**Reproducibility.** To ensure the reproducibility of this paper, we have included a schematic overview of the JOSENet framework in Fig. 1, a high-level structure of our VICReg solution in Fig. 2, and a configuration paragraph at section 3.2 that gives additional information about the expanders dimensions. Also, in Section 4.1, datasets, preprocessing methods and data augmentation used in our work are presented. Moreover, the hyperparameters used in the primary supervised model and in the auxiliary SSL models are listed in section 4.2 and 4.3, respectively. In Section A.2.1 of the appendix, we added the architecture details about the FGN together with its block diagram in Fig. 3 and its parameters in Table 9. Moreover, additional details about the adaptations of the other SSL architectures can be found in both sections A.3 and A.4. Lastly, we provide the source code and the instructions to reproduce our experiments. The code is based on Python and Pytorch framework. Unless otherwise stated, we have used an NVIDIA RTX5000 Quadro GPU with 16GB GDDR6 and 8 virtual cores by Intel Xeon 4215 CPU with a frequency of 3.2 GHz. The code is available at https://anonymous.4open.science/r/JOSENet.

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

## A  APPENDIX

This appendix includes additional materials to the main paper. Section A.1 contains an overview of both violence and action recognition datasets analyzed. Section A.2 provides a more detailed description of the architecture utilized in the primary task, together with an explanation of the importance of the spatial dropout technique, while Section A.3 describes further adaptation details of the self-supervised techniques. The section A.4 describes additional implementation details utilized to compare JOSENet with state-of-the-art methods. Finally, section A.5 gives additional details on the Siamese architecture while section A.6 shows results obtained by using our framework for the action recognition task and the performance on an independent test set.

### A.1  DATASET ANALYSIS

### A.1.1  VIOLENCE DETECTION DATASETS

In the last years, several violence detection datasets have been deployed to deal with this task. In 2011, the first two datasets for violence detection were released: the Hockey Fight Dataset Bermejo Nievas et al. (2011) which collects 1000 clips from actions of the National Hockey League, and the Movie Fight Dataset Bermejo Nievas et al. (2011) made of 200 clips taken from a variety of action movies. Actually, the first dataset lacks diversity since all the videos represent the same subject and environment, while the second dataset is small for usage in the present day. A year

Table 6: Comparison between the various violence datasets. The "natural" scenario means that the videos are realistic but recorded by different types of devices (e.g. mobile cameras).

| Dataset | # Videos | Clip (sec) | Resolution | Annotation | Scenario |
|---|---|---|---|---|---|
| Hockey Fight Bermejo Nievas et al. (2011) | 1,000 | 1.6-1.96 | 360x288 | Video | Hockey |
| Movies Fight Bermejo Nievas et al. (2011) | 200 | 1.6-2 | 720x480 | Video | Movie |
| Crowd Violence Hassner et al. (2012) | 246 | 1.04-6.52 | Variable | Video | Natural |
| UCF-Crime Sultani et al. (2019) | 1,900 | 60-600 | Variable | Video | CCTV |
| CCTV-Fights Perez et al. (2019) | 1,000 | 5-720 | Variable | Frame | Natural |
| Surv-Fight Akti et al. (2019) | 300 | 2 | Variable | Video | CCTV |
| RWF-2000 Cheng et al. (2021) | 2,000 | 5 | Variable | Video | CCTV |
| UBI-Fight Degardin & Proenca (2020) | 1,000 | 0-600 | 640x360 | Frame | Natural |
| AIRTLab Sernani et al. (2021) | 350 | 2-14 | 1920x1080 | Video | Artificial |

Table 7: Comparison between the various action recognition datasets.

| Dataset | # Videos | Clip (sec) | Actions | Clips (per cat.) |
|---|---|---|---|---|
| HMDB51 Kuehne et al. (2011) | 6,766 | $\sim 3$ | 51 | min. 101 |
| UCF101 Soomro et al. (2012) | 13,320 | 1-71 | 101 | min. 102 |
| Kinetics400 Kay et al. (2017) | 306,245 | 10 | 400 | min. 400 |

later, the Crowd Violence Dataset Hassner et al. (2012) was assembled and became the first dataset that considers the usage of real-world surveillance footage. Besides its novelty and its challenging overcrowded scenes, the quality of the videos is poor. More recently, Sultani et al. (2019) presents the UCF-Crime dataset composed of 128 hours of real-world surveillance videos. The videos are labeled with 13 anomalies such as fighting, burglary, road accidents, explosions, etc. One of the latest datasets focused on real-world fights in surveillance videos is the CCTV-Fights Dataset Perez et al. (2019) which contains 1000 videos with about 8 hours of frame-level annotations. Despite its name, only 280 videos are from surveillance cameras. One of its greater disadvantages is the unbalance of videos and frames between fight and non-fight; 216 fight videos are present and the number of fight frames represents only 3% of the entire dataset. A dataset worth mentioning is the Surveillance Camera Fight Dataset Akti et al. (2019) which contains only 300 videos, equally distributed between fight and non-fight. Even if the number of samples is small and the quality of the videos is not high, the dataset contains only surveillance videos with different scenarios, light, and color conditions. An important dataset for violence detection is the UBI-Fight dataset Degardin & Proenca (2020) which consists of around 80 hours of video fully annotated at frame level. Unfortunately, it is not focused on CCTV videos, in fact most of them are recorded with camera movements which are not feasible for our scenario. The RWF-2000 dataset Cheng et al. (2021) has been released in order to cope with the issues that exist with the previous datasets. It is made of 2,000 videos, 5 seconds long, and captured at 30 FPS by real-world surveillance cameras. Finally, in the last year, the AIRTLab dataset Sernani et al. (2021) has been created with the purpose of testing the robustness of violence detection techniques to false positives: hugs, claps, high-fives, etc. All the videos are recorded in natural lighting conditions and in the same room. A comparison of all the datasets taken into consideration is shown in Table 6.

Table 8: Datasets utilized in target and auxiliary tasks. The number of segments generated for the training and validation sets can be found in the last two columns (assuming $N = 16$ as the number of frames per segment, with $N_s = 7.5$ FPS).

| Dataset | Task | # Train | # Validation |
|---|---|---|---|
| RWF-2000 Cheng et al. (2021) | Target | 3,200 | 800 |
| HMDB51 Kuehne et al. (2011) | Auxiliary | 3,873 | 1,590 |
| UCF101 Soomro et al. (2012) | Auxiliary | 27,418 | 10,697 |
| UCF-Crime Sultani et al. (2019) | Auxiliary | 29,149 | 4,814 |

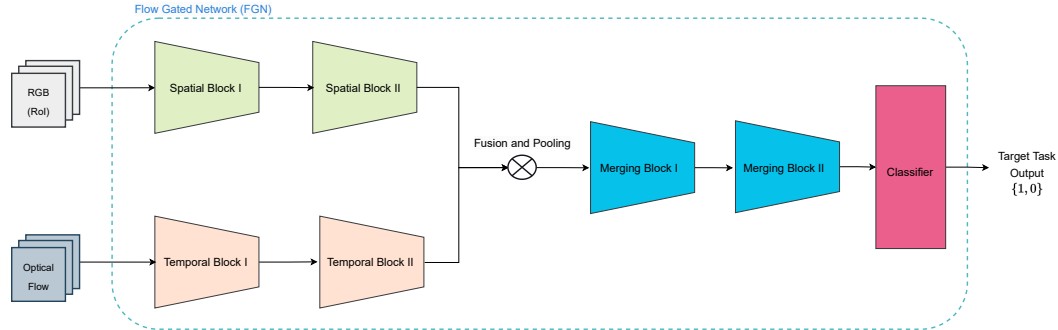

Figure 3: The architecture of the Flow Gated Network (FGN) is utilized as a primary supervised model for violence detection. A segment of length $N = 16$ with $N_s = 7.5$ (FPS) is received as input. The Spatial Block receives as input RGB frames cropped to obtain the Region of Interest (RoI), while the Temporal Block is fed with optical flow frames.

Table 9: Parameters of the proposed FGN architecture. "Type" indicates the category of layers utilized in that particular block. "Filter Shape" shows the dimension of the filter applied in that particular layer (None if that layer does not require any filter). "T" represents the number of repeats of that particular block.

| Block Name | Type | Filter Shape | T |
|---|---|---|---|
| Spatial/Temporal Block I | Conv3D+BatchNorm3D | $1 \times 3 \times 3$ @ 16 | 2 |
| | Conv3D+BatchNorm3D | $3 \times 1 \times 1$ @ 16 | |
| | MaxPool3D | $1 \times 2 \times 2$ | |
| | Dropout3D | None | |
| Spatial/Temporal Block II | Conv3D+BatchNorm3D | $1 \times 3 \times 3$ @ 32 | 2 |
| | Conv3D+BatchNorm3D | $3 \times 1 \times 1$ @ 32 | |
| | MaxPool3D | $1 \times 2 \times 2$ | |
| | Dropout3D | None | |
| Fusion and Pooling | Multiply | None | 1 |
| | MaxPool3D | $8 \times 1 \times 1$ | |
| Merging Block I | Conv3D+BatchNorm3D | $1 \times 3 \times 3$ @ 64 | 2 |
| | Conv3D+BatchNorm3D | $3 \times 1 \times 1$ @ 64 | |
| | MaxPool3D | $1 \times 2 \times 2$ | |
| Merging Block II | Conv3D+BatchNorm3D | $1 \times 3 \times 3$ @ 128 | 1 |
| | Conv3D+BatchNorm3D | $3 \times 1 \times 1$ @ 128 | |
| | MaxPool3D | $2 \times 3 \times 3$ | |
| Classifier | FC Layer | 128 | 2 |
| | Sigmoid | 1 | 1 |

A.1.2 ACTION RECOGNITION DATASETS

Since violence detection is a subclass of the action recognition task, it is crucial to explore the possibilities offered by the research in this field. Indeed, most self-supervised learning (SSL) techniques rely on a few relevant action recognition datasets. Before 2011, the human action recognition datasets contained only around ten different action categories. To solve this problem, in that year the HMDB51 dataset Kuehne et al. (2011) was released, containing 51 classes spread over 6,766 clips. Shortly after, a larger dataset called UCF101 Soomro et al. (2012) has been created. It has nearly twice the size of HMDB51 with regards to clips and the number of labels. All the videos are taken from the internet and recorded in an unconstrained environment with camera motion, partial occlusion, low-quality frames, etc. These datasets are not large enough to train the latest action recognition deep learning models; this is the reason why DeepMind has created the Kinetic dataset Kay et al. (2017). In its first version, it is comprehensive of more than 300 thousand clips and

Table 10: Hyperparameters tuning results. The first row indicates the results obtained by using a starting set of parameters with a default value. Then, by using an incremental approach, we obtain in the last row the results for our baseline model.

| Parameters | Value | Accuracy | F1 | TNR | TPR | AUC |
|---|---|---|---|---|---|---|
| Initial | Default | 74.44 | 72.42 | 74 | 76 | 0.778 |
| Augmentation | True | 79.72 | 79.69 | 84 | 76 | 0.882 |
| $N_s$ | 7.5 | 84 | 84 | 86 | 81 | 0.9 |
| Batch Size | 32 | 84.25 | 84.24 | 85 | 83.5 | 0.916 |
| Spatial Dropout | 0.2 | 85.87 | 85.87 | 85.5 | 86.25 | 0.924 |

400 different actions. In Table 7, we summed up the characteristics of the main action recognition datasets.

Finally, Table 8 shows the number of segments produced for the training and validation sets of each dataset selected for auxiliary (HDMB51, UCF101, UCF-Crime) and target task (RWF-2000).

## A.2 PRIMARY NETWORK ARCHITECTURE DETAILS

### A.2.1 THE TWO-STREAM FLOW-GATED NETWORK

The two-stream Flow-Gated Network (FGN) architecture for the primary task is made of three main blocks: the spatial block, the temporal block and the merging block. The features extracted from both spatial and temporal blocks are fused via multiplication and pooled by a 3D max pooling layer. Since the output of the sigmoid varies between 0 and 1, and given that the max-pooling can reserve local maximum, the output of the RGB channel multiplied by 1 has a larger probability to be retained, while an output multiplied by 0 is probably dropped. For this reason, this kind of architecture is called "Flow Gated Network": it implements a self-learned pooling strategy that utilizes the temporal block as a gate. Then, the merging block aims to enhance the chosen features, which are fed as input into a classifier that is responsible for the classification part.

Notice that, regarding the architecture proposed in Cheng et al. (2021), a slight modification of the max pooling layers of the merging block is applied to work with smaller segments. Moreover, batch normalization layers are adopted to prevent vanishing gradients and spatial dropout is used to reduce overfitting. ReLU activation functions are used after each convolutional layer to increase the nonlinearities, except for the last two Conv3D layers of the temporal block where sigmoids are applied. In figure 3 the FGN architecture is presented, while in Table 9, a detailed description of each block is shown.

### A.2.2 HYPERPARAMETERS TUNING

To build the baseline model, we made several trials to find the best combination of hyperparameters. Using an incremental approach, we started our experiments with a default configuration of parameters: dropout $p = 0.2$, SGD optimizer, momentum 0.9, weight decay 1e-6, $N = 16$, and 30 epochs with patience of 15. At this point, no augmentation or spatial dropout was applied, the $N_s$ was set to 12.8 FPS and batch size to 16. In the first row of Table 10 we have shown the results on this first trial. To initially tackle the overfitting problem, we applied some data augmentation techniques such as color jitter and random flip, as described in section A.3.3. Then, with the aim of reducing the computational cost, we tried different frame rates and we found that using a reduced $N_s = 7.5$ FPS we also obtained a strong improvement in performance. In addition, we increased the batch size to 32 which further improved the results. Finally, since our model still suffers from overfitting, we implemented the spatial dropout technique as explained in section A.2.3.

### A.2.3 BENEFITS OF THE SPATIAL DROPOUT ON JOSENET

During the development of the baseline model, we have observed that the network obtains good results but at the same time, it is clearly affected by overfitting. The plot in figure 4 (left) confirms the issue. As we know, dropout is one of the simplest methods to tackle this problem because it improves generalization performance and prevents the activations from being strongly correlated. Instead of using the classical dropout technique in the case of 2D or 3D architectures, it is more beneficial to use the spatial dropout Tompson et al. (2015). Differently from the standard dropout

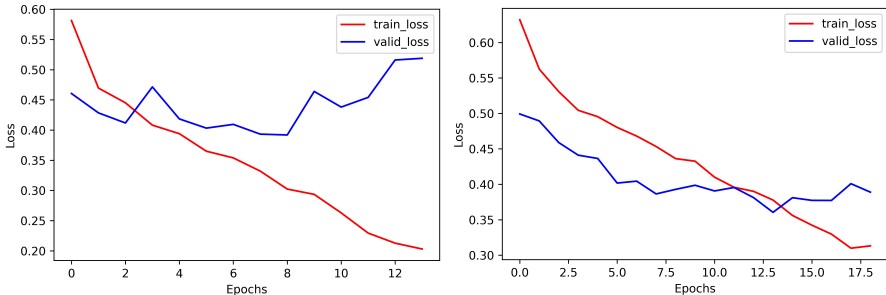

Figure 4: Comparison between the loss functions with (left) and without (right) the spatial dropout technique. As we can see, the spatial dropout successfully reduces overfitting compared to the previous model.

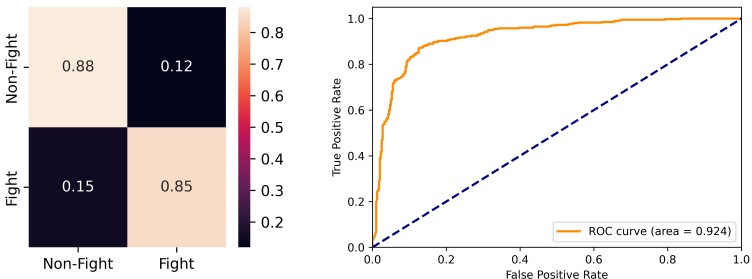

Figure 5: Normalized confusion matrix (left) and ROC curve (right) obtained by evaluating our best model on the RWF-2000 validation set by pretraining via our VICReg proposed method with 64 batch size on the entire UCF-Crime dataset. The rows and columns of the confusion matrix represent respectively the predicted and target labels.

which randomly excludes with probability $p$ some convolution feature map activation (a pixel), the spatial dropout produces a random zero-out of an entire feature map. This is very useful and proved to be more efficient than a standard dropout. We find that applying the spatial dropout with 0.2 probability as the last step of the first two 3D convolutions (in both spatial and temporal blocks) significantly improves the performance. The benefit of the spatial dropout can be seen in the plot of figure 4 (right) where the overfitting is reduced considerably.

The spatial dropout, together with the pretraining technique developed in our work has allowed us to further reduce the overfitting and to reach excellent performance. In fact, instead of using a transfer learning strategy where the model could risk overfitting on labels Yang et al. (2020), our JOSENet framework is based on SSL that uses supervisory signals generated from the data itself. These results are confirmed in figure 5 where the confusion matrix and the ROC curve of the best model are shown.

### A.3 ADAPTATIONS OF SELF-SUPERVISED ARCHITECTURES TO THE VIOLENCE DETECTION PROBLEM

#### A.3.1 ARCHITECTURES

In our paper, we tested and compared several SSL methods for the addressed problem of violence detection. However, each SSL technique was originally developed and tested by using a particular architecture, which on some occasions differs greatly from our FGN. For this reason, some modifications are necessary on the FGN blocks in order to develop a viable pretraining for our architecture. We believe that slightly modifying each architecture to make it as similar as possible to the target architecture is crucial to correctly exploit each SSL method.

To implement the odd-one-out (O3D) method we pretrained 6 Siamese RGB branches. At the end of each branch, we applied an average pooling with kernel size $9 \times 7 \times 7$ to obtain a flattened embedding of size 128 that is used in the fusion layer. The arrow-of-time (AoT) technique relies on optical flow input which is a segment of 32 optical flow frames split into two parts and fed as input in the respective Siamese network. Two Siamese optical flow branches are used so that the pretrained weights can be used in the target task. As in the reference paper, the output of these branches is concatenated and is fed as input into a series of 3 convolutional layers, each one followed by batch normalization. We used for each convolutional layer a kernel size of $(8, 7, 7)$ with stride and padding $(1, 1, 1)$ and finally a Global Average pooling layer (GAP) of size $(1, 2, 2)$. All these dimensions are chosen and tested to obtain the best performance and produce a consistent output dimension.

The only method that applies 3D convolutional neural networks (CNNs) is the space-time cubic puzzle (STCP) (i.e., using a 4-tower Siamese network of 3D ResNets). We decide to simply substitute the 3D ResNet architecture with our spatial block. To concatenate the output of each branch to obtain an array of size 2048, we choose to use an average pooling layer of size $4 \times 2 \times 2$ at the end of each block. Thus a fully-connected layer is applied with an output size of 24, which is equal to the number of classes to be predicted from the network. The resulting number of classes is $4! = 24$, due to the choice of avoiding flipping upside-down the videos.

### A.3.2 PREPROCESSING

In the AoT task, to avoid trivial learning (by adding a zero flow frame at the end of the segment), a total of $32 + 1$ RGB frames are used to generate 32 optical flow frames that are processed by the AoT network. For the STCP task, we sample a number of $N = 64$ frames, removing the temporal jittering in STCP auxiliary task. Indeed, we observe that by implementing temporal jittering (i.e., using $N = 128$) the dataset would be too small to produce useful features. Moreover, to prevent a slow convergence during training, we avoid the upside-down flip suggested in Kim et al. (2019). Thus, the total number of classes for the STCP task is $c = 24$. Lastly, both O3D and VICReg receive as input 16 frames as the target task.

### A.3.3 DATA AUGMENTATION

In the target task, we apply color jitter to RGB frames and random flip to both RGB and flow frames. The color jitter randomly changes the brightness, contrast, saturation, and hue of an image with a range of $[-0.2, 0.2]$. Both color jitter and random flip are applied with a probability of 50%. To maintain coherence between optical flow and RGB frames, the random flip is applied to both of them. After the data augmentation (in both target and auxiliary tasks), a standardization is applied per input vector, as in Cheng et al. (2021). On the other hand, for each auxiliary task, we use different kinds of augmentations following the reference papers. When no augmentation is applied, we keep consistent with the data augmentation techniques used in the target task. In STCP, apart from channel replication and random jitter, no additional augmentation is necessary Kim et al. (2019). Since the VICReg pretext task is a regularized method that needs to learn representations that are invariant to different distortions, it is crucial to produce the right amount of data augmentation. The RGB augmentation pipeline follows the protocol of Bardes et al. (2021); Chen et al. (2020). Differently from Bardes et al. (2021); Chen et al. (2020), we use a stronger random cropping strategy in the RGB augmentation pipeline, by imposing a scaling factor within the range $[0.08, 0.1]$, which is confirmed to be more efficient from our tests. We call this augmentation the "zoom crop" strategy. In other words, similarly to Bardes et al. (2022), we want to exploit local features produced by a cropped view of the RGB frames. For what concerns the flow segments, a simple horizontal flip (with 50% probability) is applied. This approach seems to be coherent with the target task learned features. In fact, the RGB branch of the FGN receives as input a cropped version of the RGB segment (ROI), while the flow segment remains unchanged. A visualization of the VICReg video input can be seen on the left side of figure 2 of the paper.

### A.3.4 VICREG PRETRAINING ADDITIONAL DETAILS

The auxiliary SSL VICReg model has been trained using the loss in equation 1 that consists of three different terms:

- the **variance regularization term** $v$ can be seen as a hinge function and it is computed along the batch dimension as the standard deviation of the embeddings:

$$v(Z) = \frac{1}{d} \sum_{j=1}^{d} \max(0, \gamma - S(z^j, \epsilon)), \tag{2}$$

where $S$ is the regularized standard deviation $S(x, \epsilon) = \sqrt{Var(x) + \epsilon}$. In particular, $\gamma$ is a target for the regularized standard deviation that is generally fixed to 1, while $\epsilon$ is a small value that prevent numerical instabilities. This term aims to prevent a complete collapse of the embeddings because it promotes the variance of the batch to be equal to $\gamma$. It is necessary to use the standard deviation instead of the variance to avoid a vanishing gradient problem that would produce collapsed embeddings.

- the **covariance regularization term** $s$

$$c(Z) = \frac{1}{d} \sum_{i \neq j} [C(Z)]_{i,j}^2 \tag{3}$$

is the sum of the squared off-diagonal coefficients of the covariance matrix $C(Z)$ scaled by a factor of $\frac{1}{d}$, such that

$$C(Z) = \frac{1}{n-1} \sum_{i=1}^{n} (z_i - \bar{z})(z_i - \bar{z})^T, \text{where } \bar{z} = \frac{1}{n} \sum_{i=1}^{n} z_i \tag{4}$$

This term pushes the off-diagonal coefficients of $C(Z)$ to be zero. This means that the network is encouraged to decorrelate the dimensions of the embeddings so that similar information is not encoded. It seems that this kind of decorrelation at the embedding level leads to a decorrelation effect also at the representation level.

- the **invariance term** $s$ between $Z$ and $Z'$ is

$$s(Z, Z') = \frac{1}{n} \sum_i ||z_i - z'_i||^2 \tag{5}$$

which is the mean-squared euclidean distance between each pair of embedding vectors.

We tested several weight term combination ($\lambda$, $\mu$, and $\nu$) but we have found that the values suggested by Bardes et al. (2021) were the best configuration also in our scenario. The training of the Auxiliary SSL VICreg model has been done for 100 epochs with patience of 15. We have used a SGD optimizer, weight decay 1e-6, momentum 0.9, and learning rate of 1e-2. A cosine annealing scheduler is also applied with maximum number of iterations equal to the number of epochs.

## A.4 STATE-OF-THE-ART IMPLEMENTATION DETAILS

We compared JOSENet with 3 different state-of-the-art SSL methods by pretraining them on a 15% subset of UCF101. For all these methods we followed the experiments made by Han et al. (2020). However, since we used a reduced dataset, for a fair comparison we decided to reduce accordingly some of the hyper-parameters of each method. For InfoNCE Oord et al. (2019) we pretrained separately both FGN RGB $f_\theta^{InfoNCE}$ and optical flow branch $f_{\theta'}^{'InfoNCE}$ for 45 epochs. The same number of epochs and modalities are used for UberNCE Han et al. (2020), producing both $f_\theta^{UberNCE}$ and $f_{\theta'}^{'UberNCE}$. Instead, the initialization stage of CoCLR Han et al. (2020) is done by utilizing the same weights produced during InfoNCE pretraining. Then, the alternation stage proceeds by first training the FGN RGB branch $f_\theta^{CoCLR}$ for 15 epochs and using the optical flow branch $f_{\theta'}^{'InfoNCE}$ to mine hard positive samples. Finally, the alternation stage concludes by training the FGN optical flow branch $f_{\theta'}^{'CoCLR}$ for 15 epochs, by using as pretraining $f_\theta^{CoCLR}$ for the RGB branch together with $f_{\theta'}^{'InfoNCE}$ for optical flow branch weights. For all these methods, all the learning rate schedules are reduced accordingly and a down-sampling rate of 3 (which corresponds to 7.5 FPS) is used to maintain consistency with all the other experiments.

Table 11: Action recognition results of our FGN pretrained with VICReg on UCF-Crime. The last two columns represent respectively the Top-1 and Top-5 accuracy computed on fold 1 for both datasets.

| Dataset | VICReg | Top-1 | Top-5 |
|---------|--------|-------|-------|
| HMDB51  | ✗      | 20    | 49.03 |
| HMDB51  | ✓      | 24.62 | 55.03 |
| UCF101  | ✗      | 43.18 | 70.12 |
| UCF101  | ✓      | 48.1  | 73.24 |

### A.5 Siamese architecture for VICReg Auxiliary Model

As with the previous techniques (i.e. O3D, AoT, and STCP), we develop two novel Siamese structures for both RGB branch and the optical flow branch. In particular, two Siamese RGB/flow blocks $f_\theta$ receive as input respectively an augmented version of an RGB/flow segment. These networks produce the representations $Y$ and $Y'$ and, in this case, an adaptive max-pooling layer of size $4 \times 4 \times 4$ is applied to both of them, reducing the representation dimension to 2048 (as in Bardes et al. (2021)). These resulting vectors are the input of the expanders $h_\phi$.

### A.6 Additional experiments

#### A.6.1 Generalizing JOSENet to Action Recognition Tasks

As an additional test, we use our JOSENet framework pretrained with VICReg on the UCF-Crime dataset as a starting point for the action recognition task. In particular, we fine-tune the FGN on two different benchmark datasets for action recognition: HMDB51 and UCF101.

In order to avoid any overfitting of the network on dynamic actions, the only modification that we apply is the removal of the RoI extraction for the RGB frames. The entire training procedure and the architecture remain unchanged compared to the one described in the previous sections.

The results in Table 11 show that our pretraining based on the VICReg seems to be effective also for a more generic action recognition task. Indeed, in both the datasets we obtain a boost in performances of about 3-6% of accuracy. We want to highlight that our architecture trained from scratch surpasses by 1-3% Top-1 accuracy of the Resnet3D-18 Hara et al. (2018), which requires a huge computational power (33.3M parameters) compared to the FGN network (272,690 parameters). Thus, we strongly believe our JOSENet framework could obtain near state-of-the-art performance by exploiting a pretraining obtained on a larger dataset (i.e. Kinetics).

#### A.6.2 Results on an Independent Dataset.

With the aim to have an insight into the generalization error, we further test our model on an independent test set. One of the few candidates available for this task is the Surveillance Camera Fight dataset Akti et al. (2019), which is composed of 300 videos taken from surveillance cameras. We surpass the state-of-the-art results on the selected dataset with an accuracy of 80.8% (+8.8%). We have to notice that the Surveillance Camera Fight presents a small portion of videos with a quality that varies widely. Thus, we can assert that testing our model on a better and larger test set would improve the understanding of the generalization capability of the model.

#### A.6.3 Performance Evolution

To enhance the understanding of the paper, in this section we outlined the JOSENet performance evolution described in section ??. The baseline (first row in Table 12) is the first model obtained and does not involve any pretraining: the primary task has been addressed directly by the FGN trained on the RWF-2000 training set. Then, we implemented several non-regularized SSL strategies that produce slightly improved results, compared to the baseline. For this reason, we used VICReg as an auxiliary SSL model in our JOSENet framework. With the aim of selecting the best pretraining dataset for JOSENet we have seen that UCF-Crime obtained the higher results (see second row

Table 12: The evolution of the results of our JOSENet framework. In the first row the results of the baseline model are shown. The final results are shown in the last row.

| Dataset | Accuracy | F1 | TNR | TPR | AUC |
|---|---|---|---|---|---|
| No-pretraining | 85.87 | 85.87 | 85.5 | 86.25 | 0.924 |
| UCF-Crime (15%) | **86.75** | **86.74** | 84.5 | **89** | 0.918 |
| UCF-Crime (100%) | 86.5 | 86.5 | **88** | 85 | **0.924** |

of Table 12). Finally, we pretrained the auxiliary SSL VICReg model on the entire UCF-Crime, reaching the final results for JOSENet, shown in the last row of Table 12.

