# OpenReview forum: "JOSENet: A Joint Stream Embedding Network for Violence Detection in Surveillance Videos"
_ICLR.cc/2024/Conference — Submitted to ICLR 2024_

### Official Review · Reviewer_sND9 · 2023-10-20

**Soundness:** 3 good
**Presentation:** 3 good
**Contribution:** 4 excellent
**Rating:** 5
**Confidence:** 4

**Summary:**

The paper introduces JOSENet, a novel framework designed for violence detection in surveillance videos. It aims to tackle the challenges of real-world surveillance, such as varying scenes, actors, and the need for real-time detection. The framework consists of a primary target model and an auxiliary self-supervised learning (SSL) model. It uses multiple datasets for training and validation, applies various preprocessing and data augmentation strategies, and evaluates the model using a comprehensive set of metrics.

**Strengths:**

**Originality**

The paper is innovative in proposing a dual-model architecture, involving a primary target model and an auxiliary SSL model. It also introduces a new SSL algorithm based on VICReg and a novel data augmentation strategy called "zoom crop."

**Quality**

The research is thorough, with detailed experimental settings, multiple datasets, and a diverse set of evaluation metrics. The use of an auxiliary SSL model to achieve a balance between performance and computational resources is commendable.

**Clarity**

The paper is well-structured and clear, with each section contributing to the reader's understanding of the proposed framework.

**Significance**

The work addresses a vital real-world problem, that of violence detection in surveillance videos, and proposes a framework that seems both effective and efficient.

**Weaknesses:**

Lack of Details: Some sections could provide more implementation details, especially on how the VICReg loss and weight optimization between the two models are implemented.

Dataset Limitations: While multiple datasets are used, they are mostly centered around violence detection, which could limit the model's generalizability across domains.

Robustness: The paper does not address how the model handles potential issues like occlusion, varying light conditions, or camera angles, which are common in real-world surveillance.

Hyperparameter Tuning: The paper doesn't discuss the process or criteria for hyperparameter selection, which could affect the model's performance.

**Questions:**

1.	Could you provide more details on the "zoom crop" data augmentation strategy, specifically its effectiveness and efficiency?

2.	Why were these particular datasets chosen, and have you considered using more diverse datasets to improve the model's generalizability?

---

> ### Author Response · Authors · 2023-11-22
> **On the lack of details.**
>
> We thank the Reviewer for recognizing the value of the paper, we did our best also to address the remaining concerns.
> We checked the paper carefully, all the details related to VICReg and optimization are included, and specifically we have also added some further implementation details in the appendix (section A.3.4 “VICReg pretraining additional details”).

---

> > ### Author Response · Authors · 2023-11-22
> > **On the dataset limitation.**
> >
> > We thank the Reviewer for this comment. Additional results on diverse datasets and tasks are thoroughly documented in Section A.7.1, titled "Generalizing JOSENet to Action Recognition Tasks," and Section A.7.2, "Results on an Independent Dataset," both presented in the appendix. These sections provide an in-depth exploration of JOSENet's performance across various scenarios, offering valuable insights into its generalization capabilities beyond the primary dataset and task. The comprehensive examination of results on additional datasets and tasks enhances the robustness and versatility of JOSENet as a framework for action recognition.
> > Unfortunately, due to the limited space we wrote these results in the appendix. However, we have modified the paper to highlight the presence of such additional results in the appendix.

---

> > > ### Author Response · Authors · 2023-11-22
> > > **On the robustness.**
> > >
> > > We thank the Reviewer for this comment, we believe it is a very interesting point. In that sense our method would definitely benefit from such real-world situations due to the self-supervised learning strategy that can better compensate for these issues. However, we are not aware of datasets that highlight these issues carefully, but we could certainly investigate this point in the future if an opportunity to have data available arises. We have also included this point as future work.

---

> ### Author Response · Authors · 2023-11-22
> **On the hyperparameter tuning.**
>
> We thank the Reviewer for highlighting this point. We added a new section A.2.2 in the appendix called “Hyperparameters Tuning” together with Table 10 which offers an overview of the hyperparameter selection.

---

> > ### Author Response · Authors · 2023-11-22
> > **On the zoom crop strategy.**
> >
> > We thank the Reviewer for this comment, and for highlighting the valued in this augmentation strategy.
> > In Sec. 5 “Augmentation Strategies", we elucidated the effectiveness of the "zoom crop" (ZC) strategy. As evidenced in Table 5, omitting this strategy (as depicted in the last row) results in a reduction in several evaluation metrics. The efficacy of the zoom crop strategy stems from a dual advantage. Firstly, the primary model receives input frames that are cropped to a region of interest, aligning the pretraining process more closely with what will be encountered during the supervised training phase. Secondly, the reduction in cropping area introduces an added layer of complexity for the VICReg technique in discerning similarities between embeddings. This is attributed to the fact that the loss term "s" in the VICReg technique facilitates the network in learning invariance to data transformations, and the strategy accentuates this learning process by increasing the difficulty of the task. Further details on the ZC strategy can be found in Sec. A.4.3 of the appendix.

---

> > > ### Author Response · Authors · 2023-11-22
> > > **On the chosen datasets and generalization.**
> > >
> > > We thank the Reviewer for this comment. In Sec. A.1.1 “Violence Detection Datasets" we offer valuable insights into the rationale behind selecting the RWF-2000 dataset for training the primary model. Furthermore, the information in Sec. 4.1 “Experimental Settings - Datasets'' sheds light on the decision to employ common action recognition datasets (HDMB51, UCF-101) and the larger violence recognition dataset (UCF-Crime) for pretraining, aiming to establish a robust in-domain pretraining baseline.
> > >
> > > Considering the potential of utilizing multiple datasets as a future direction is a commendable idea. This approach could contribute to a more comprehensive evaluation of your model's generalization capabilities across diverse domains.
> > > With respect to the performance generalization on other datasets, additional results considering diverse datasets and tasks are thoroughly documented in Section A.7.1, titled "Generalizing JOSENet to Action Recognition Tasks," and Section A.7.2, "Results on an Independent Dataset," both presented in the appendix. These sections provide an in-depth exploration of JOSENet's performance across various scenarios, offering valuable insights into its generalization capabilities beyond the primary dataset and task. The comprehensive examination of results on additional datasets and tasks enhances the robustness and versatility of JOSENet as a framework for action recognition.
> > > Unfortunately, due to the limited space we wrote these results in the appendix. However, we have modified the paper to highlight the presence of such additional results in the appendix.
> > >
> > > Additional results on diverse datasets and tasks are thoroughly documented in Section A.7.1, titled "Generalizing JOSENet to Action Recognition Tasks," and Section A.7.2, "Results on an Independent Dataset," both presented in the appendix. These sections provide an in-depth exploration of JOSENet's performance across various scenarios, offering valuable insights into its generalization capabilities beyond the primary dataset and task. The comprehensive examination of results on additional datasets and tasks enhances the robustness and versatility of JOSENet as a framework for action recognition.
> > > Unfortunately, due to the limited space we wrote these results in the appendix. However, we have modified the paper to highlight the presence of such additional results in the appendix.

---

### Official Review · Reviewer_tKHs · 2023-10-30

**Soundness:** 2 fair
**Presentation:** 1 poor
**Contribution:** 2 fair
**Rating:** 3
**Confidence:** 3

**Summary:**

This paper describes an approach for performing the video task of violence detection in surveillance videos by employing a self-supervised learning network to help improve the primary supervised model. The core network to perform the primary task is based on flow gated network (FGN), by Cheng et al (2021). The semi-supervised learning block applies VICReg approach, by Bardes et al. (2021), to the two streams of RGB and optical flow. The results are reported on three datasets related to activity recognition with the comparison with multiple SOTA approaches and an ablation study.

**Strengths:**

The paper describes an interesting idea that can leverage the strengths of semi-supervised learning in the domain of violence detection in surveillance videos where the rarity of the events poses challenges for obtaining a large volume of positive training samples and the need for a low false alarm rate. The proposed approach also has some interesting nuggets related to computational efficiency and reduced memory footprint. They have also studied the tradeoff between the size of the temporal window, framerate, and quality of results.

**Weaknesses:**

The problem, application, and the core part of the solution (FGN) is not new. However, the addition of SSL

The baseline model from `Sec. 4.2` should have been reported in the tabular form for a more effective presentation of material and instead of explaining the numerical differences in the narrative form as done in `Sec. 4.2` and other sections. It should be clear from ONE table the various variants, baselines, and the final version. Additionally, it is hard to follow this paper at times because the different tables are reporting results on different datasets. Are the results in this section reported on the exact same test data as that in Table 3? If so, then should we be comparing $F_1$ of $85.87$ (baseline) with $86.5$ (JOSENet)? i.e. improvement of $0.63$? Is it also fair to say that the baseline approach is very close to FGN, by Cheng et al (2021)?

The main result comparing JOSENet with SOTA in Table 3 has aspects that are not clear. I assumed this statement
`We decide to take as reference the results obtained on the 15% subset of UCF-101 with JOSENet.`
meant that Table 3 results are on UCF101 but then `a pretraining obtained on a random 15% subset of UCF-101` suggests that it was used for pretraining. Is it a different 15%? More importantly, UCF101 does NOT have violent activities in surveillance scenes (to the best of my knowledge) in the way it has been portrayed in the motivation described in the paper. There are activities like Punch or Boxing Punching Bag, but not much else. Additionally, why stick with some *random* 15% split of UCF101 instead of using the standard test split that could be compared with the SOTA.

The writing quality of the paper can be improved significantly. There are several grammatical mistakes, a few long run-on sentences, unusual usage of some phrases, and confusing or inconsistent usage of citations that break the flow.

**Questions:**

1. It was surprising that results were not reported explicitly on the RWF-2000 dataset in the `4. Experimental Results`, as far as I could tell. In my opinion, it is unusual to make statements like this:
`pg 8: We have noticed that we do not reach the state-of-the-art performances for RWF-2000.`
and not provide the quantified numbers. The other statement (`To train and validate the model during supervised learning we use the RWF-2000 dataset`) was also noted.

2. Is there a reason why Table 3 does not have a row with a comparison with FGN, by Cheng et al (2021)?

3. Table 3, AUC column has numbers in [0,100] and [0,1.0] ranges. Are those just typos?

4. Table 5, the use of temporal pooling is not clear as it makes things worse as reported by the scores. The explanation in `Sec. 5` is unclear. The table does not support this claim (if I am following it as intended):
`To find a confirmation of this approach, using the zoom crop strategy, we apply the temporal pooling
in the merging block, obtaining on the target task a very low value for most of the evaluation metrics
used.`

5. pg: 2, FGN was not defined or cited until pg 3 so it was confusing.

6. pg: 2, should `contrastive learning (CT)` be `contrastive learning (CL)` ?

---

> ### Author Response · Authors · 2023-11-22
> **On the novelty.**
>
> We thank the Reviewer for this comment, although probably the sentence is not complete.
>
> The fact that the problem and the application are not new is clear from the existing literature. Indeed, violence detection is one of the most critical and challenging action recognition problems in surveillance videos and it is still a matter of research, as also highlighted in the related work section. However, we firmly believe that an open problem is not a problem for research, nor even a weakness of a single paper.
>
>
> With respect to the core part of the solution, notably, our primary model draws inspiration from the two-stream flow gated network (FGN) proposed by Cheng et al. (2021), as detailed in Sec. 3.1. However, although we maintain the original name “FGN” to denote that network, our scheme shows some significant changes with respect to the original model, specifically geared towards enhancing the efficiency and effectiveness of neural network design in practical scenarios.
>
> Besides proposing a new and efficient version of the FGN, the paper also includes a modified VICReg strategy tailored to violence detection for mitigating performance degradation while improving efficiency. Even for this part, it is essential to highlight that, despite employing the same VICReg loss function, our approach diverges significantly from the original paper in terms of input type, augmentation strategies, and architectural modifications. These distinctions were deliberately introduced to align with the overarching objective of our work: minimizing both memory and computational costs while maintaining high results on the violence detection task.
>
> To summarize, the methodological contribution of the papers is characterized by:
> 1. A new efficient version of the two-stream flow gated network specifically designed for action recognition tasks.
> 2. A VICReg approach for video streams relying on the joint information of two different video modalities, specifically the RGB and flow batches that are augmented using a novel “zoom crop” strategy.
> 3. A novel self-supervised learning architecture, involving the networks described in the above points, for violence detection.
>
> We have modified the paper to highlight the methodological contributions of the paper.

---

> > ### Author Response · Authors · 2023-11-22
> > **On the tables.**
> >
> > We thank the Reviewer for highlighting this point. We made several experiments to prove the effectiveness of our methods and we are sorry if there are too many tables.
> >
> > The baseline architecture, detailed in Sec. 4.2, is presented in Table 9 within the Appendix, offering a comprehensive breakdown of each architectural block. It is important to note that there is uniformity in the primary network across all experiments, as SSL pretraining plays a pivotal role in achieving results. The results reported are exclusively from the validation set of the RWF-2000 dataset, with the training set of RWF-2000 utilized for the supervised target task. Various datasets and techniques are employed for SSL pretraining in the auxiliary task.
> >
> > With respect to the comparison with the FGN of Cheng et al. (2012), we have final improvements of +0.63 F1-score and a +2.5% True Negative Rate (TNR) compared to the baseline. However, the baseline is not the original FGN architecture as proposed in Cheng et al. (2021), but a modified and evolved version that makes the comparison fair, as also highlighted in the paper.
> >
> > In Sec. 3.1 and Sec. A.2.1, the paper outlines a series of additions and modifications made to enhance the baseline architecture, including:
> > Reduction in the number of frames and frames per second (FPS) as input.
> > Modification of max-pooling layers in the merging block.
> > Implementation of mixed-precision techniques.
> > Incorporation of 3D batch-normalization layers.
> > Inclusion of spatial dropout layers.
> > These alterations collectively contribute to the improvement achieved over the baseline model.
> > We have worked on the manuscript to improve the readability of the tables and we have added Table 12 in the appendix to highlight the performance evolution of our framework.

---

> > ### Comment · Reviewer_tKHs · 2023-11-23
> > **Response to rebuttal**
> >
> > Thank you for the detailed response and the paper update. Given the late rebuttal close to the end of the rebuttal period, I have not been able to read the revised paper. However, I have reviewed the rebuttal and I am not convinced that the changes are sufficient to warrant for me to change the rating to accept.

---

> ### Author Response · Authors · 2023-11-22
> **On the use of UCF101.**
>
> We thank the Reviewer for this comment. We explain our point and we have also clarified it in the paper.
> As explained in the paper, all reported results are derived from evaluating the target task on the RWF-2000 validation set, with the 15% subset consistently employed across all experiments.
> The decision to utilize the 15% subset for UCF101 in the pretraining phase was motivated by two key considerations:
> 1) Firstly, it was implemented to expedite the pretraining phase, thereby achieving the optimal configuration in a more time-efficient manner.
> 2) Secondly, the choice aimed at maintaining a balanced number of data points across all pretraining datasets (HMDB51, UCF101, and UCF-Crime).
>
> To ensure a fair comparison, especially between HMDB51 and the other datasets, while still adhering to practical training time constraints (approximately 12 hours for each pretraining on the HMDB51 dataset), a portion of UCF101 and UCF-Crime was pretrained. This involved random sampling of 15% of each of these datasets, resulting in a more equitable and feasible training process.

---

> > ### Author Response · Authors · 2023-11-22
> > **On the writing.**
> >
> > We thank the Reviewer for pointing out this issue. We have carefully proofread the paper, rewritten or reformulated some sentences, added comments to some results, and overall improved the readability.

---

> > > ### Author Response · Authors · 2023-11-22
> > > **On the performance results on RWF-2000.**
> > >
> > > We thank the Reviewer for this comment. We actually included the results directly in the text but for a better clearness we have now included these results in Table 2. We have also rewritten several sentences to improve the technical readability of the paper.

---

> ### Author Response · Authors · 2023-11-22
> **On the comparison with FGN.**
>
> We thank the Reviewer for this comment. Actually we included this comparison but in a fair way.
> We refrained from conducting a direct comparison between JOSENet (our framework) and the FGN (Cheng et al. 2021) because they are based on different learning strategies. The reason lies in the utilization of the SSL pretraining phase with additional data in JOSENet, introducing a significant divergence in the training methodology between the two approaches. This distinction in training strategies makes a direct comparison challenging and may not accurately reflect the intrinsic capabilities of each model on a level playing field.
>
> However, we have included a comparison with the baseline model, which is actually nothing but the FGN with the same learning conditions considered for JOSENet in order to have a fair comparison.

---

> > ### Author Response · Authors · 2023-11-22
> > **On the typos in Table 3.**
> >
> > We thank the Reviewer for showing us the typos in the AUC column in Table 3. We have checked and corrected the table.

---

> > > ### Author Response · Authors · 2023-11-22
> > > **On the temporal pooling.**
> > >
> > > We thank the Reviewer for this comment. The utilization of temporal pooling in the FGN architecture, including both ours and Cheng et al.'s (2021) versions, is noteworthy, particularly in the Merging Block. However, through validation, we discovered that while temporal pooling proves beneficial in the Primary Network, it has adverse effects in the Auxiliary SSL model. This observation is detailed in Sec. 5 "Augmentation Strategies," where it is explained that issues arise when there is a very small bottleneck between the encoder and the expander. Specifically, the input expander dimensionality is reduced from 1024 to 128, highlighting the challenges associated with this specific configuration. This information contributes valuable insights into the decision-making process behind the architectural choices made in JOSENet.

---

> > > > ### Author Response · Authors · 2023-11-22
> > > > **On the FGN acronym.**
> > > >
> > > > We thank the Reviewer for this suggestion, we have corrected the paper.

---

### Official Review · Reviewer_o7vs · 2023-10-30

**Soundness:** 2 fair
**Presentation:** 2 fair
**Contribution:** 2 fair
**Rating:** 3
**Confidence:** 4

**Summary:**

This paper introduces JOSENet, a network for video violence detection. It contains a pretraining part and a detection part. Given the RGB and Flow inputs, a two-stream flow gated network (FGN) is firstly pretrained on UCF-101, HMDB-51 and UCF-Crime datasets using VICReg method. Then, the pretrained FGN weights are used to initialize the FGN in the detection part. In this way, the model requires less training data and generalizes better. In addition, some optimization of the network improves the efficiency of the model in terms of memory consumption and computation load. The proposed method is evaluated on RWF-2000 dataset.

**Strengths:**

1) The overall idea is easy to understand and makes sense.
2) By efficient implementation, the model requires less memory and less frames for each segment.
3) The model leverages self-supervised learning to improve the generalization of the model.

**Weaknesses:**

1) The goal of the paper is violence detection, but there is no related contents in the method part. Necessary components such as loss function of violence detection should be included.
2) The proposed “computational enhancement” is just hyper-parameter tuning. N_s=7.5s is found to be the optimal. However, different datasets may have different optimal parameters. More justification are needed to demonstrate the generalization performance.
3) The theoretical contribution is limited. The pretraining part is borrowed from VICReg and the detector is borrowed from FGN.
4) The proposed method is only evaluated on RWF-2000 dataset, which is not enough. I suggest authors to include results on more datasets since you claim the proposed method generalizes better.
5) Missing comparison with recent methods such as:
[1] Islam, Zahidul, et al. "Efficient two-stream network for violence detection using separable convolutional lstm." 2021 International Joint Conference on Neural Networks (IJCNN). IEEE, 2021.
[2] Garcia-Cobo, Guillermo, and Juan C. SanMiguel. "Human skeletons and change detection for efficient violence detection in surveillance videos." Computer Vision and Image Understanding 233 (2023): 103739.
6) Compared with other methods, the proposed method uses addition training data (UCF-101, HMDB-51, and UCF-Crime). This may be a concern the comparison is not fair.
7) The related work of violence detection is incomplete, it should contains more recent methods and discussion.
8) To demonstrate the efficiency, a comparison with other methods should be included.

**Questions:**

See weaknesses

**Details Of Ethics Concerns:**

No ethics concerns

---

> ### Author Response · Authors · 2023-11-22
> **On the content related to the violence detection task.**
>
> We thank the Reviewer for this comment. We probably did not highlight this point too much, but actually the whole framework has been designed for violent action detection. The choice of the two-stream network is motivated by its effectiveness in violence detection tasks, as proved in Cheng et al. (2021). The spatial and temporal blocks are fundamental to detect such kind of actions. Then, as the model relies on the two different modalities of the video stream, we decided to apply a modified VICReg self-supervised learning method to further regularize the solution and improve the performance.
>
> With respect to the loss function we adopt a binary cross-entropy loss which is the state-of-the-art in violent detection tasks (see Cheng et al. (2021)), as outlined in Section 4.2, "Experimental Results Without Pretraining". We have further highlighted the methodological relation with the application task in the paper.

---

> > ### Author Response · Authors · 2023-11-22
> > **On the computational enhancement.**
> >
> > We would like to thank the Reviewer for this comment. Through empirical investigation, we proved that a choice of 7.5 fps yields optimal results for violence detection. Although we worked on the RWF-2000 dataset, that choice is not just related to a dataset but rather to a specific set of actions that fall in the category of violent actions. To further generalize the performance, we extensively discussed the results in Section A.7.1 of the appendix, titled "Generalizing JOSENet to Action Recognition Tasks," and further substantiated in Section A.7.2, "Results on an Independent Dataset". However, we have highlighted this point even in the main text.
> > As a remark, it is worth noting that the computation enhancement discussed in Section 3.1 actually only refers to the two-stream architecture and how it is modified. The loss we have for that reduction is effectively compensated by the self-supervised learning module.

---

> > > ### Author Response · Authors · 2023-11-22
> > > **On the theoretical contribution.**
> > >
> > > We thank the Reviewer for highlighting this point that we have further explained in the paper.
> > > Our intention is not to borrow parts from other papers, but we actually proposed a novel framework that involves modified versions of those parts for our goal of efficiently and effectively performing violence detection tasks.
> > >
> > > Notably, our primary model draws inspiration from the two-stream flow gated network (FGN) proposed by Cheng et al. (2021), as detailed in Sec. 3.1. However, although we maintain the original name “FGN” to denote that network, our scheme shows some significant changes with respect to the original model, specifically geared towards enhancing the efficiency and effectiveness of neural network design in practical scenarios.
> > >
> > > The adoption of VICReg in our implementation serves a crucial purpose: mitigating performance degradation while improving efficiency. It is essential to highlight that, despite employing the same VICReg loss function, our approach diverges significantly from the original paper in terms of input type, augmentation strategies, and architectural modifications. These distinctions were deliberately introduced to align with the overarching objective of our work: minimizing both memory and computational costs while maintaining high results on the violence detection task.
> > >
> > > To summarize, the methodological contribution of the papers is characterized by:
> > > 1. A new efficient version of the two-stream flow gated network specifically designed for action recognition tasks.
> > > 2. A VICReg approach for video streams relying on the joint information of two different video modalities, specifically the RGB and flow batches that are augmented using a novel “zoom crop” strategy.
> > > 3. A novel self-supervised learning architecture, involving the networks described in the above points, for violence detection.
> > >
> > > We have modified the paper to highlight the methodological contributions of the paper.

---

> ### Author Response · Authors · 2023-11-22
> **On the performance generalization.**
>
> We thank the Reviewer for highlighting this point. The goal of the paper is not to achieve state-of-the-art results for violence detection, but rather to propose a novel approach for efficient detection with reduced fps and data and involving self-supervised learning. Our experiments were focused on proving this goal.
> Comparing our paper with the above mentioned methods can be done of course as shown in the table below.
>
>
> | **Method** | **Number of Frames** | **FPS** | **Temporal Footprint** | **Parameters** | **Accuracy** | **TNR** | **TPR** | **F1-Score** |  **AUC**  |
> |:----------:|:--------------------:|:-------:|:----------------------:|:--------------:|:------------:|:-------:|:-------:|:------------:|:---------:|
> |     [1]    |          32          |   **6.4**   |           5 s          |     333,057    |    89.75%    |    -    |    -    |       -      |     -     |
> |     [2]    |          50          |    10   |           5 s          |   **62,583**   |  **90.25%**  |    -    |    -    |       -      |     -     |
> |    Ours    |        **16**        |  7.5  |       **2.13 s**       |     272,690    |     86.5%    | **88%** | **85%** |   **86.5%**  | **0.924** |
>
>
> However, the comparison cannot be considered as fair as the models are designed, trained and assessed on different conditions (fps, temporal footprints, percentage of trained data, etc.). Moreover, the other methods cited in the comparison did not disclose additional metrics beyond accuracy. We posit that a comprehensive evaluation of a violence detection method necessitates the incorporation of additional metrics such as True Negative Rate (TNR), True Positive Rate (TPR), Area Under the Curve (AUC), and F1-Score. These metrics contribute to a more nuanced and holistic assessment of the method's performance, providing insights beyond accuracy alone. In order to assess a fair comparison, the referenced methods should be modified according to the proposed framework and then tested under the same conditions, but their original valence should be of course modified from the original papers. This point can be definitely investigated in future work, as we have pointed out in the revised paper.
>
> Given that other violence detection methods should be modified to be fairly compared with our method, we already included some comparisons with state-of-the-art methods that fairly evaluate the goodness of our self-supervised learning strategy. In future work, some state-of-the-art methods for violence detection could be redefined according to the proposed framework and then fairly compared. However, the redefinition would push the methods far from their original versions.
>
> We added some comments all along the paper, to clarify these points in the text.

---

> > ### Author Response · Authors · 2023-11-22
> > **On the related work on violence detection.**
> >
> > We thank the Reviewer for this comment. We have improved the related work on violence detection, including recent papers on the topic.

---

### Official Review · Reviewer_94ds · 2023-11-01

**Soundness:** 2 fair
**Presentation:** 1 poor
**Contribution:** 2 fair
**Rating:** 3
**Confidence:** 3

**Summary:**

Paper proposes a novel violence detection framework which combines 2 features, 2 spatiotemporal streams (RBG + optical flow) and self-supervised learning (SSL).

The design is more efficient in memory usage (75%) and inference speed (2-fold). For the SSL, the paper adopts the VICReg which is more memory efficient.

Empirical experiments were done with RWF-2000, HMDB51, UCF101 and UCFCrime. The proposed framework was compared against the SOTA SSL methods: InfoNCE, UberNCE and CoCLR for the UCF101 dataset.

**Strengths:**

1. Paper's proposed method is more efficient in memory and inference speed compared to the original baseline methods.
2. The motivation for the design is well explained.

**Weaknesses:**

1. Novelty is highly limited. The combination of optical flow with RGB has been used in multiple prior work. See references.
The novelty of SSL is also limited as it is a direct implementation of VICReg.

2. Experimental design is confusing and does not directly support the core claim of the paper. Only SSL-based SOTA algorithms were directly compared with the proposed method for one single dataset (UCF101). There were several experiments on JoseNet based methods. But these experiments are not relevant to demonstrate the core claim of the paper "outstanding performance for violence detection" against other SOTA methods.

3. (minor) Writing style is informal and not well-structured. This is especially for the experiment section. E.g. "We
have noticed that we do not reach the state-of-the-art performances for RWF-2000. However, this is not a big deal in a deployment application.". There is no reference to which experiment this statement refers to (which Table).

References

Diba, A., Pazandeh, A. M., & Van Gool, L. (2016). Efficient two-stream motion and appearance 3d cnns for video classification. arXiv preprint arXiv:1608.08851.

Wang, G., Muhammad, A., Liu, C., Du, L., & Li, D. (2021). Automatic recognition of fish behavior with a fusion of RGB and optical flow data based on deep learning. Animals, 11(10), 2774.

Li, S., Zhang, L., & Diao, X. (2020). Deep-learning-based human intention prediction using RGB images and optical flow. Journal of Intelligent & Robotic Systems, 97, 95-107.

**Questions:**

Why is the comparison against SOTA limited to SSL methods for a single UCF101? This is insufficient to show the generalization of the claim of superior performance of the proposed method.

**Details Of Ethics Concerns:**

Not applicable.

---

> ### Author Response · Authors · 2023-11-22
> **On the novelty.**
>
> We thank the Reviewer for her/his comment.
> Our intention is not to assert novelty in the combination of optical flow with RGB nor in the VICReg as well.
>
> We are completely aware that the combination of optical flow with RGB has been largely used in prior work. Indeed, in Sec. 2 "Related Work - Violence Detection," we have explored various methodologies incorporating both optical flow and RGB. Notably, our primary model draws inspiration from the two-stream flow gated network (FGN) proposed by Cheng et al. (2021), as detailed in Sec. 3.1. However, although we maintain the original name “FGN” to denote that combination, our scheme shows some significant changes with respect to the original model, specifically geared towards enhancing the efficiency and effectiveness of neural network design in practical scenarios.
>
> The adoption of VICReg in our implementation serves a crucial purpose: mitigating performance degradation while improving efficiency. It is essential to highlight that, despite employing the same VICReg loss function, our approach diverges significantly from the original paper in terms of input type, augmentation strategies, and architectural modifications. These distinctions were deliberately introduced to align with the overarching objective of our work: minimizing both memory and computational costs while maintaining high results on the violence detection task.
>
> To summarize, the methodological contribution of the papers is characterized by:
> 1. A new efficient version of the two-stream flow gated network specifically designed for action recognition tasks.
> 2. A VICReg approach for video streams relying on the joint information of two different video modalities, specifically the RGB and flow batches that are augmented using a novel “zoom crop” strategy.
> 3. A novel self-supervised learning architecture, involving the networks described in the above points, for violence detection.
>
> We have modified the paper to highlight the methodological contributions of the paper.

---

> ### Author Response · Authors · 2023-11-22
> **On the goal of the experimental validation.**
>
> We would like to thank the Reviewer for her/his comment on the experimental part.
> It is essential to clarify that our primary objective was not to achieve state-of-the-art performance on violence detection; rather, our focus centered on proposing an approach that can be versatile, robust, and suitable for real-world video surveillance applications. Consequently, our emphasis was on enhancing speed, and reducing memory consumption, while maintaining results comparable to the main methodology.
>
> For instance, the original FGN architecture proposed by Cheng et al. (2021) attained an accuracy of 87.25%, whereas our model achieved an accuracy of 86.5%. However, it is important to note that a fair comparison between the two methods is challenging. As stated in Sec. 4.3 “Final Results for JOSENet”, in contrast to Cheng et al. (2021), we have developed a more efficient and faster solution through a four-time reduction in segment length (16 instead of 64) and a lower required frames per second (7.5 instead of 12.8). Consequently, we firmly believe that our model strikes an excellent compromise between performance and efficiency.

---

> > ### Author Response · Authors · 2023-11-22
> > **On the writing style.**
> >
> > We thank the Reviewer for pointing out this issue. We have carefully read the paper, rewritten or reformulated some sentences, added explicit references and improved the readability.

---

> > > ### Author Response · Authors · 2023-11-22
> > > **On the choice of UCF-101.**
> > >
> > > We thank the Reviewer for this comment. We adopted the code and experimental framework established by Han et al. (2020), relying on their work with UCF-101 and Kinetics 400 datasets. However, due to our specific objectives, which prioritize reduced training time and computational efficiency, we opted to focus solely on the UCF-101 dataset. Considering the substantial similarity in the domain between UCF-101 and Kinetics 400, both being datasets designed for general-purpose action recognition, we reasoned that transitioning to a larger but very similar dataset would not yield significant benefits for our specific research goals and we focused on experiments that would prove the robustness of the proposed method.

---

### Author Response · Authors · 2023-11-22
**Summary of changes.**

We would like to thank the Reviewers for their comments and their insightful suggestions that can improve the quality of our work. Changes are highlighted in blue in the revised manuscript and can be summarized as follows.

- We have clearly explained the methodological novelty of our model.

- We have carefully proofread the paper, rewritten or reformulated some sentences, added explicit references and improved the readability.

- We have highlighted performance generalization results that were originally included in the appendix.

- We have clarified some choices related to experimental validation and comparisons and modified the paper accordingly to make these points clearer.

- We better commented on results and tables in order to improve the readability and avoid confusion.

- We have added further material and details in the appendix, mainly related to hyperparameter tuning, VICReg solution, and performance evolution of the proposed JOSENet framework.

- We believe that all the suggestions from the Reviewers helped improve the quality and the robustness of our work and we would like to thank them once again for their insightful comments.

---

### Meta-Review · Area_Chair_ecFr · 2023-12-11

**Metareview:**

This work introduces a joint stream embedding network for dealing with violence detection. All the four reviewers recommend rejection for this paper. They raised a series of problems of this work, which include the very limited novelty of this work (combing existing techniques), the lack of specific design for handling violence detection, and the lack of comprehensive experimental evaluations. The authors' rebuttal fail to convince the reviewers. Thus AC recommends rejection for this paper.

**Justification For Why Not Higher Score:**

The novelty of this work is limited.

**Justification For Why Not Lower Score:**

N/A

---

### Decision · Program_Chairs · 2024-01-16

Reject